# Sex-Based Clinical Outcome in Advanced NSCLC Patients Undergoing PD-1/PD-L1 Inhibitor Therapy—A Retrospective Bi-Centric Cohort Study

**DOI:** 10.3390/cancers14010093

**Published:** 2021-12-24

**Authors:** David Lang, Anna Brauner, Florian Huemer, Gabriel Rinnerthaler, Andreas Horner, Romana Wass, Elmar Brehm, Bernhard Kaiser, Richard Greil, Bernd Lamprecht

**Affiliations:** 1Department of Pulmonology, Johannes Kepler University Hospital Linz, Krankenhausstrasse 9, 4020 Linz, Austria; andreas.horner@kepleruniklinikum.at (A.H.); romana.wass@kepleruniklinikum.at (R.W.); elmar.brehm@kepleruniklinikum.at (E.B.); bernhardkaiser@gmx.at (B.K.); bernd.lamprecht@kepleruniklinikum.at (B.L.); 2Medical Faculty, Johannes Kepler University Linz, Altenberger Strasse 69, 4020 Linz, Austria; a.brauner@gmx.at; 3Oncologic Center, Department of Internal Medicine III with Haematology, Medical Oncology, Haemostaseology, Infectiology and Rheumatology, Paracelsus Medical University, 5020 Salzburg, Austria; f.huemer@salk.at (F.H.); g.rinnerthaler@salk.at (G.R.); r.greil@salk.at (R.G.); 4Cancer Cluster Salzburg, 5020 Salzburg, Austria; 5Salzburg Cancer Research Institute-Laboratory for Immunological and Molecular Cancer Research (SCRI-LIMCR), 5020 Salzburg, Austria

**Keywords:** immunotherapy, immune-checkpoint inhibitor, response prediction, men and women, pembrolizumab, nivolumab, atezolizumab, ECOG, CRP, chemo-immunotherapy

## Abstract

**Simple Summary:**

Retrospective analyses suggest that men treated with immune-checkpoint inhibitor (ICI) monotherapy for non-small cell lung cancer (NSCLC) have better outcomes than women. However, female patients have more favorable responses when chemotherapy (CHT) is given together with ICI. We aimed to explore the clinical impact of such sex differences in two cohorts, receiving ICI monotherapy or ICI-CHT combination, respectively. We found no significant difference in outcomes between men and women treated with either therapeutic regimen. However, known predictive factors for ICI response such as the expression of programmed-death ligand 1 (PD-L1) on tumor cells or patient performance status had significant implications for men rather than for women. Our results warrant increased research efforts to clarify sex-specific differences in anti-tumor immune response mechanisms and in the efficacy of ICI therapies, especially in women.

**Abstract:**

Men with non-small cell lung cancer (NSCLC) have a more favorable response to immune-checkpoint inhibitor (ICI) monotherapy, while women especially benefit from ICI-chemotherapy (CHT) combinations. To elucidate such sex differences in clinical practice, we retrospectively analyzed two cohorts treated with either ICI monotherapy (*n* = 228) or ICI-CHT combination treatment (*n* = 80) for advanced NSCLC. Kaplan–Meier analyses were used to calculate progression-free (PFS) and overall survival (OS), influencing variables were evaluated using Cox-regression analyses. No significant sex differences for PFS/OS could be detected in either cohort. Men receiving ICI monotherapy had a statistically significant independent impact on PFS by Eastern Cooperative Oncology Group performance status (ECOG) ≥2 (hazard ratio (HR) 1.90, 95% confidence interval (CI): 1.10–3.29, *p* = 0.021), higher C-reactive protein (CRP; HR 1.06, 95%CI: 1.00–1.11, *p* = 0.037) and negative programmed death-ligand 1 (PD-L1) status (HR 2.04, 95%CI: 1.32–3.15, *p* = 0.001), and on OS by CRP (HR 1.09, 95%CI: 1.03–1.14, *p* = 0.002). In men on ICI-CHT combinations, multivariate analyses (MVA) revealed squamous histology (HR 4.00, 95%CI: 1.41–11.2, *p* = 0.009) significant for PFS; and ECOG ≥ 2 (HR 5.58, 95%CI: 1.88–16.5, *p* = 0.002) and CRP (HR 1.19, 95%CI: 1.06–1.32, *p* = 0.002) for OS. Among women undergoing ICI monotherapy, no variable proved significant for PFS, while ECOG ≥ 2 had a significant interaction with OS (HR 1.90, 95%CI 1.04–3.46, *p* = 0.037). Women treated with ICI-CHT had significant MVA findings for CRP with both PFS (HR 1.09, 95%CI: 1.02–1.16, *p* = 0.007) and OS (HR 1.11, 95%CI: 1.03–1.19, *p* = 0.004). Although men and women responded similarly to both ICI mono- and ICI-CHT treatment, predictors of response differed by sex.

## 1. Introduction

Cancer immunotherapy using immune-checkpoint inhibitors (ICI) directed against programmed death-ligand 1 (PD-L1) or programmed cell death protein 1 (PD-1) has revolutionized lung cancer treatment [1]. Originally established as monotherapy for pretreated advanced non-small cell lung cancer (NSCLC) and first-line therapy in patients with high PD-L1 expression [2,3,4,5,6], its fields of application have continued to expand. In stage IV disease without targetable genetic alteration and PD-L1 expression <50%, first-line ICI-chemotherapy (CHT) combination treatment has become state-of-the art [7,8,9], while the additional value of the combination of nivolumab and the Cytotoxic T-lymphocyte-associated Protein 4 (CTLA-4) inhibitor ipilimumab with or without CHT is yet to be determined [10,11].

Early phase-3 ICI monotherapy studies in NSCLC, with the exception of the OAK study for atezolizumab, showed a numerical prognostic benefit favoring men in subgroup analyses [2,3,4,5,6]. This finding was initially confirmed in a systematic review and meta-analysis by Conforti et al. for NSCLC as well as for melanoma [12]. However, a subsequently published similar investigation by Wallis et al. reported no such sex difference [13]. In the ICI-CHT combination setting in NSCLC, a systematic review and meta-analyses showed a larger benefit for women as compared to men [14]. Possible explanations for these observed sex differences may be variations in the composition of the tumor microenvironment, in T-cell differentiation as well as in mechanisms of tumor immune evasion [15].

Our study group has evaluated various biomarkers for lung cancer immunotherapy in real-life cohorts, where women repeatedly had more favorable hazard ratios for progression-free survival (PFS) and overall survival (OS) as compared to men [16,17,18]. Although these differences have been small and mostly non-significant, such findings seemed interesting in the light of large-scale metanalyses demonstrating the opposite. In the present study we thus aimed to evaluate whether such sex differences could be detected in our recently updated bi-centric ICI monotherapy database as well as in a cohort of ICI-CHT-treated patients. For both cohorts, we additionally intended to elucidate sex-specific differences in the prognostic relevance of various patient- or tumor-related factors.

## 2. Materials and Methods

The bi-centric ICI monotherapy cohort consisted of 228 retrospectively evaluated consecutive patients with advanced NSCLC that had received at least one cycle of either nivolumab, pembrolizumab or atezolizumab at the lung cancer unit of Kepler University Hospital Linz or at the Medical Oncology unit of Paracelsus Medical University Salzburg between May 2015 and December 2019. The ICI-CHT cohort comprised 80 consecutive patients treated with platinum-based doublet CHT combined with pembrolizumab or atezolizumab (plus bevacizumab) at Kepler University Hospital Linz between June 2018 and December 2019. The patient registry as well as the present evaluation have been approved by the ethics committee of the federal state of Upper-Austria (EK Nr. 1139/2019).

Patients were retrospectively followed from ICI therapy initiation on to death or censored at the date of last verified contact. Disease progression was retrospectively defined by imaging and death, as well as by reviewing the relevant medical records. Therapy could be applied beyond disease progression in selected cases of considerable clinical benefit. In addition, in stage IV patients with PD-L1 expression <50% and contraindications to CHT, first-line ICI treatment could be initiated upon tumor-board decision. Therapy line was defined as treatment for non-curable (e.g., stage IV [19] or not otherwise treatable stage III) disease, whereas previous therapies in potentially curable stages were not considered. We excluded patients in clinical trials or on ICI/ICI combinations and patients, who had previously received an ICI therapy.

Chemo-immunotherapy was applied according to the respective phase 3 studies, using carboplatin/pemetrexed/pembrolizumab for non-squamous and carboplatin/paclitaxel/pembrolizumab for squamous-cell carcinomas [7,8]. A minority of patients received the IMpower-150 regimen of carboplatin/paclitaxel/atezolizumab/bevacizumab [9]. Cisplatin was not used, and ICI-CHT combination was routinely given for four cycles if tolerated. In maintenance therapy for patients having responded to ICI-CHT treatment, only the respective ICI substance but no CHT was continued. An earlier switch to maintenance mono-immunotherapy could be performed if the treating physician deemed the continuation of combination therapy inappropriate due to intolerance and/or toxicity.

Radiological response to ICI monotherapy was routinely assessed by a chest- and upper abdomen CT scan using iodinated contrast medium every 10 to 12 weeks, equaling four cycles of nivolumab or three cycles of pembrolizumab or atezolizumab. In the ICI-CHT cohort, equivalent re-staging was performed after every two cycles of combination therapy. Re-staging could be preponed due to suspected disease progression and imaging modalities such as ^18^F-FDG-PET/CT or cerebral magnetic resonance tomography could be additionally conducted according to the treating clinician’s judgement.

All statistical analyses were accomplished using R (R: A Language and Environment for Statistical Computing; Version 4.1.1). Sex-specific differences in baseline characteristics were tested for statistical significance using a two-samples *t*-test or the Mann-Whitney-U-test for non-normally distributed variables; categorical variables were tested using the Chi–Square–Test. Kaplan–Meier-analyses were used to calculate PFS and OS in all patients as well as according to sex. Results were expressed as median in months (95% confidence interval (CI)) unless otherwise specified. The Kaplan–Meier curves were compared statistically using the log rank test, whereas a *p*-value < 0.05 was regarded statistically significant. Uni- and multivariate models for predictive factors of PFS and OS in the ICI monotherapy and ICI-CHT cohort in all patients and according to sex were accomplished using Cox-regression analyses. Variables included in these models were age (years), sex (only in the models for all patients), smoking status (< vs. ≥5 pack years), histological subtype (adeno- vs. squamous-cell carcinoma), palliative therapy line (1,2 vs. ≥3; only in the models for ICI monotherapy), Eastern Cooperative Oncology Group performance status (ECOG; 0.1 vs. ≥2) and presence of a targetable genetic tumor alteration (anaplastic lymphoma kinase (ALK), epidermal growth factor receptor (EGFR), proto-oncogene tyrosine-protein kinase ROS (ROS-1)). We also included C-reactive protein and absolute lymphocyte count, assessed using a Cobas^®^ 8000 modular analyzer (Roche Diagnostics International AG, Rotkreuz, Switzerland), and a Sysmex^®^ XN-3000 hematology analyzer (Sysmex Europe GmbH, Norderstedt, Germany), respectively. Furthermore, all multivariate models comprised PD-L1 expression on tumor cells determined using a 22C3 assay for Autostainer Link 48 by Dako (Agilent Technologies, Santa Clara, CA, USA), a negative PD-L1 status was defined as membranous staining on <1% of viable tumor cells. Regardless of their statistical significance, ECOG and PD-L1 were included into all multivariate models for PFS/OS due to their well-established predictive implications, using the “augmented backward elimination package” in R.

## 3. Results

Baseline patient and tumor characteristics of the ICI monotherapy and ICI-CHT combination cohort for all patients and separately for men and women are shown in Table 1.

Kaplan–Meier analyses (Figure 1) showed no significant sex difference, neither in the ICI monotherapy cohort (PFS: men 3M (3–5), women 3M (3–6), *p* = 0.273; OS: men 10M (8–14), women 10M (6–14), *p* = 0.592), nor in the CHT-ICI cohort (PFS: men 6M (5–10), women 5M (3-/), *p* = 0.780; OS: men 15M (10-NA), OS women: 10M (7-NA), *p* = 0.399), respectively.

Uni- and multivariate predictors of PFS and OS for both therapy cohorts are presented in Table 2.

In the ICI monotherapy cohort, multivariate analyses showed ECOG and PD-L1 status as significant predictors of PFS, while OS was significantly influenced by ECOG and CRP. Chemo-immunotherapy-treated patients displayed a significant multivariate influence of ECOG, CRP and PD-L1 on PFS and of ECOG, presence of a targetable genetic tumor alteration and CRP on OS. Sex did not play a significant role in these analyses; neither alone, nor in combination with any other variable evaluated.

Uni- and multivariate analyses for PFS and OS according to sex are shown in Table 3 and Table 4.

Among women treated with ICI monotherapy, no significant predictor of PFS could be identified, while ECOG was significant for OS. Female patients in the ICI-CHT cohort showed a significant interaction of both PFS and OS with CRP. Men in the ICI monotherapy cohort had a significant impact of ECOG, CRP and PD-L1 status on PFS and of CRP on OS. Among male patients in the ICI-CHT cohort, squamous histology predicted reduced PFS, while ECOG and CRP proved significant for OS.

## 4. Discussion

Our data suggest that sex did not significantly influence outcomes in the reported cohorts of patients having received PD-1/PD-L1 directed ICI therapy either alone or in combination with platinum-based doublet CHT.

In the ICI monotherapy setting, multivariate models revealed ECOG and PD-L1 status as the main variables for PFS, while ECOG and CRP were most relevant for OS. Similarly, in the ICI-CHT combination cohort, ECOG, CRP and PD-L1 were the main determinants of PFS, and ECOG, presence of a targetable genetic alteration and CRP of OS. These findings largely resemble the current state of knowledge on predictive biomarkers for prediction of response to ICI therapy: Despite several years of biomarker research, PD-L1 expression, performance status and presence of a targetable genetic alteration as in EGFR or ALK are still the most common biomarkers routinely applied in daily clinical practice, knowing that their predictive power is limited [20,21,22]. In retrospective cohorts, elevated CRP has been demonstrated to be associated with inferior prognosis and may also positively correlate with PD-L1 expression [23,24,25]. Similarly, several other biomarkers related to the tumor microenvironment such as tumor-infiltrating lymphocytes or more readily available clinical biomarkers such as peripheral lymphocyte count or neutrophil to lymphocyte ratio have been suggested [16,20,22], but are of only limited use and thus are not directly reflected in the therapeutic decision algorithms of current guidelines [1].

Evidence from several metanalyses suggests that men have more favorable responses to ICI monotherapies, while women have a comparably larger benefit on ICI-CHT combinations [12,13]. Although we did not detect significant differences between men and women concerning PFS and OS in either reported therapy group, sex-specific uni- and multivariate analyses of predictive variables revealed interesting findings: Men had a significant impact of rather “traditional” factors such as ECOG and PD-L1 as well as CRP and squamous histology for PFS and OS in the treatment cohorts. Women on the other hand showed no significant prognostic variable for ICI monotherapy PFS and only ECOG for OS in that cohort. For both ICI-CHT PFS and OS in female patients, CRP proved significant.

Imbalances in baseline patient characteristics may have influenced these findings: Both treatment cohorts included considerably more men than women probably mainly due to the higher incidence of lung cancer among men [26], but still women have been reported less likely to receive ICI treatment for NSCLC in clinical practice [27]. This may in part be explained by the higher presence of targetable genetic alterations, especially in EGFR, among women [1,21,27,28]. In addition in our study, the prevalence of these alterations in the ICI monotherapy group was significantly higher in women, but hazard ratios for progression (PFS) and death (OS) ranged from 1.09 to 2.24 without any significant uni-and multivariate findings in both sexes and treatment modalities. Another major difference between men and women concerning baseline characteristics was an overabundance of squamous-cell carcinomas among men, which was significant in the ICI monotherapy cohort und and numerically evident in the ICI-CHT group. A higher incidence of squamous-cell carcinomas among men is well known [26,29], and most major clinical trials on ICI monotherapy and ICI-CHT combinations except for KEYNOTE-024 reported better outcomes with non-squamous histology [2,3,4,5,6,7,8]. This may have influenced our reported sex-specific outcomes, as especially in ICI-CHT combination therapy, men with squamous histology had a higher HR for PFS and OS as compared to women (2.68 and 1.93 vs. 1.04 and 1.23, respectively). Similarly, smoking history significantly differed between men and women in the ICI monotherapy cohort but did not significantly impact clinical outcomes in either sex. Recent research has shown that sex-differences in smoking habits and occupational exposure to other carcinogenic agents alone cannot fully explain the sex-specific alterations in lung cancer incidence and biology [28]. Sex hormone pathways, especially concerning β-estradiol are increasingly recognized as relevant factors in carcinogenesis and progression [28] and may be an explanation for the comparatively higher incidence of lung cancer in female never smokers and in women <50 years [28,30,31]. In our reported ICI monotherapy cohort, age differed significantly between men and women, with an overabundance of women in the group <60 and of men in the group ≥80 years; however, age did not have significant implication of PFS and OS in further analyses. 

Besides these mentioned sex differences on the demographic level, more profound variations in the pathophysiological mechanisms of anticancer immune response and tumor immune evasion contribute to the sex-specific response to ICI therapy [12,13,15,32,33]. Reportedly, women have stronger anti-tumor immune response patterns, which prompts NSCLC in female patients to develop more complex mechanisms of immune evasion, e.g., by enhanced expression of inhibitory immune checkpoint molecules leading to a higher degree of host T-cell dysfunction [15]. As a result, PD-1/PD-L1-directed monotherapy approaches may be less efficacious in women than in men. This also matches our observation that PD-L1 was shown to be a major predictor of PFS only in men receiving ICI monotherapy, although PD-L1 expression on tumor cells did not differ between sexes. 

If tumor antigenicity however can be enhanced, e.g., by the application of CHT, women may derive a larger benefit from their stronger immune responses and enhanced immune cell infiltration into the tumor [15,34]. In this regard, similar to cytotoxic agents, radiotherapy can boost antitumor immune response pathways by inducing tumor antigen release and antigen presentation [35]. Retrospective analyses among patients that had been treated with nivolumab in KEYNOTE-001 and had previously received radiotherapy showed that there was a non-significant numerical PFS and OS benefit favoring female patients [36]. In the PACIFIC trial evaluating durvalumab after chemoradiotherapy, both sexes displayed comparable PFS, but women achieved a distinctly better OS as compared to men [37].

Naturally, our study and its results are limited by its retrospective, registry-based design. Still, it reflects two cohorts of patients treated in daily clinical practice at tertiary lung cancer centers and thus allows an insight into real life data that may unearth findings not evident in clinical trial settings. Although we regard the overall sample size of the ICI monotherapy cohort rather substantial, the ICI-CHT cohort as well as several subgroups in subsequent analyses were comparably small, which confines the significance of the statistical tests reported as well as the comparability between the two therapy cohorts. In addition, reflecting the current demographics of NSCLC incidence, more men than women were included in both reported cohorts, which may limit their comparability. However, the simulation of a numerically balanced sex ratio in both therapy cohorts as described in supplementary analysis one (Appendix A) did not substantially alter PFS and OS outcomes.

## 5. Conclusions

We conclude that although there was no evident sex difference in outcomes with either mono- or chemo-immunotherapy for advanced NSCLC, prognostic factors differed between men and women. These variations may be explained by sex-specific variations in host antitumor immune response and ICI therapy effects, as well as by differences in demographic and socioeconomic factors such as smoking, comorbidities, and prevalence of targetable genetic tumor alterations. Our finding that known prognostic factors currently used in clinical practice seem to apply to male rather than to female patients urgently warrants further research in that field, specifically for women.

## Figures and Tables

**Figure 1 cancers-14-00093-f001:**
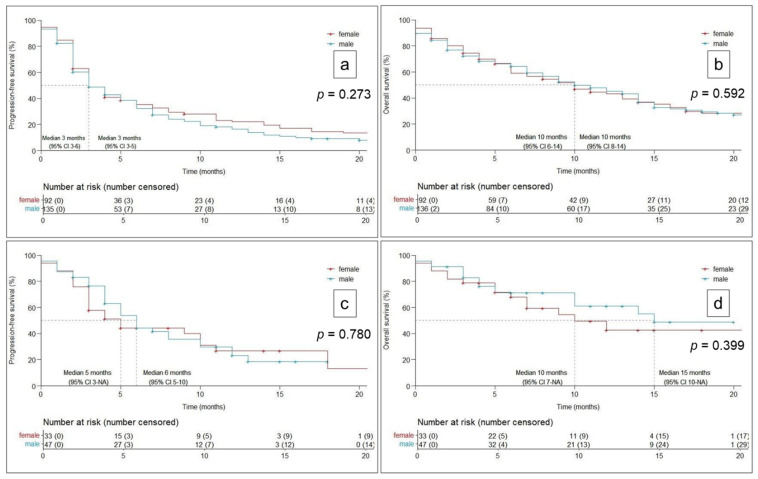
Kaplan–Meier curves for progression-free and overall survival in the mono- (**a**,**b**) and chemo-immunotherapy cohort (**c**,**d**) according to sex. Results are presented as months (95% confidence interval). CI = confidence interval, NA = not applicable.

**Table 1 cancers-14-00093-t001:** Baseline patient and tumor characteristics for all patients and according to sex in the mono- and chemo-immunotherapy cohort. Results are presented as absolute number and percent within the respective group unless otherwise specified. *p* values are for comparison between men and women.

		Mono-Immunotherapy				Chemo-Immunotherapy			
		All (*n* = 228)	Male (*n* = 136)	Female (*n* = 92)	*p*	All (*n* = 80)	Male (*n* = 47)	Female (*n* = 33)	*p*
Mean age (years; SE)		67.4 (0.71)	68.9 (1.0)	65.1 (1.0)	0.003	62.9 (1.1)	63.2 (1.3)	62.5 (1.8)	0.623
Age categories (*n*, %)					0.030				0.636
<60 years		47 (20.6)	21 (15.4)	26 (28.2)		28 (35.0)	16 (34.0)	12 (36.4)	
60–69 years		80 (35.1)	45 (33.1)	35 (38.0)		34 (42.5)	20 (42.6)	14 (42.4)	
70–79 years		78 (34.2)	53 (39.0)	25 (27.2)		17 (21.3)	11 (23.4)	6 (18.2)	
80+ years		23 (10.1)	17 (12.5)	6 (6.5)		1 (1.3)	0 (0.0)	1 (3.0)	
ECOG (*n*, %)					0.167				0.104
0.1		172 (75.4)	107 (78.7)	65 (70.7)		68 (86.1)	38 (80.9)	30 (93.8)	
≥2		56 (24.6)	29 (21.3)	27 (29.4)		11 (13.9)	9 (19.2)	2 (6.2)	
Mean pack years (SE)		45.1 (2.1)	50.9 (2.9)	36.5 (2.9)	<0.001	40.9 (3.0)	44.3 (3.7)	36.1 (4.8)	0.067
ICI substance (*n*, %)					0.897				NA
Nivolumab		90 (39.5)	52 (38.2)	38 (41.3)		-	-	-	
Pembrolizumab		105 (46.1)	64 (47.1)	41 (44.6)		77 (96.3)	45 (95.7)	32 (97)	
Atezolizumab		33 (14.5)	20 (14.7)	13 (14.1)		3 (3.8)	2 (4.3)	1 (3)	
Therapy line (*n*, %)					0.629				NA
1,2		136 (59.7)	110 (80.9)	72 (78.3)		77 (96.3)	45 (95.7)	32 (97)	
≥3		92 (40.3)	26 (21.7)	20 (21.7)		3 (3.8)	2 (4.3)	1 (3)	
Median number of ICI-CHT cycles (IQR)		-	-	-	-	4 (2)	4 (2)	4 (2)	0.962
Median number of ICI-monotherapy cycles (IQR)		4 (5)	4 (5)	4 (7)	0.343	3 (5.3)	3 (4)	2 (7)	0.761
Histological subtype (*n*, %)					0.005				0.409
Adenocarcinoma		140 (62.2)	74 (54.8)	66 (73.3)		61 (77.2)	34 (73.9)	27 (81.8)	
Squamous-cell carcinoma		85 (37.8)	61 (45.2)	24 (26.7)		18 (22.8)	12 (26.1)	6 (18.2)	
PD-L1 positive (*n*, % *)		137 (68.8)	84 (70.6)	53 (66.3)	0.517	41 (54.7)	23 (53.5)	18 (56.3)	0.812
PD-L1 expression (*n*, % **)					0.820				0.277
n.a.		29 (12.7)	18 (13.2)	11 (12)		5 (6.3)	4 (8.5)	1 (3)	
<1%		67 (29.4)	38 (27.9)	29 (31.5)		34 (42.5)	20 42.6)	14 (42.4)	
1–49%		71 (31.1)	44 (32.4)	27 (29.3)		26 (32.5)	17 (36.2)	9 (27.3)	
≥50%		61 (26.8)	36 (26.5)	25 (27.2)		15 (18.6)	6 (12.8)	9 (27.3)	
Targetable genetic alteration (*n*, %)		18 (7.9)	6 (4.4)	12 (13.0)	0.018	6 (7.5)	2 (4.3)	4 (12.1)	0.189
Mean lymphocyte count (G/L; SE)		1.4 (0.1)	1.4 (0.1)	1.4 (0.1)	0.388	1.3 (0.1)	1.3 (0.1)	1.2 (0.1)	0.973
Mean C-reactive protein (mg/dL; SE)		3.5 (0.3)	3.5 (0.4)	3.4 (0.5)	0.261	3.1 (0.6)	2.6 (0.6)	3.7 (1.2)	0.788

* Percent of patients with PD-L1 status available. ** The numeric discrepancies between PD-L1 status and PD-L1 expression are due to patients with pathologically determined positive PD-L1 status but without exact quantification being reported or with further quantification being impossible. SD = standard deviation, ECOG = Eastern Cooperative Oncology Group, ICI = Immune checkpoint inhibitor, CHT = chemotherapy, NA = not applicable, py = pack years, PD-L1 = Programmed death-ligand 1.

**Table 2 cancers-14-00093-t002:** Uni- and multivariate analyses for progression-free- and overall survival for all patients in the mono-immunotherapy and chemo-immunotherapy cohort. Results are presented as hazard ratio (95% confidence interval), with a ratio >1 signifying an increased risk of progression/death or death, respectively.

		Univariate		Multivariate		Univariate		Multivariate	
		HR (95% CI)	*p*	HR (95% CI)	*p*	HR (95% CI)	*p*	HR (95% CI)	*p*
Mono-immunotherapy (*n* = 228)		Progression-free survival				Overall survival			
Age (years)		0.99 (0.98–1.00)	0.543			0.99 (0.98–1.00)	0.753		
Female vs. male		0.86 (0.65–1.15)	0.312			0.91 (0.66–1.25)	0.551		
ECOG (≥2 vs. 0.1)		1.51 (1.09–2.10)	0.014	1.56 (1.08–2.26)	0.017	1.89 (1.31–2.74)	<0.001	1.78 (1.17–2.72)	0.008
Therapy line (≥3 vs. 1,2)		1.50 (1.08–2.10)	0.002			1.45 (1.02–2.07)	0.040		
Targetable genetic alteration (yes vs. no)		1.64 (0.99–2.70)	0.060			1.87 (1.11–3.15)	0.020		
Lymphocyte count (G/L)		0.98 (0.85–1.13)	0.783			0.87 (0.72–1.06)	0.871		
Squamous-cell vs. adenocarcinoma		0.99 (0.74–1.32)	0.917			1.18 (0.86–1.62)	0.314		
Pack years (≥5 vs. <5)		0.93 (0.60–1.44)	0.731			1.03 (0.63–1.68)	0.917		
CRP (mg/dL)		1.05 (1.01–1.08)	0.005			1.08 (1.05–1.12)	<0.001	1.06 (1.02–1.10)	0.003
PD-L1 status (neg. vs. ≥1%)		1.56 (1.13–2.14)	0.006	1.51 (1.09–2.08)	0.013	1.22 (0.86–1.73)	0.268	1.21 (0.85–1.72)	0.303
**Chemo-immunotherapy (*n* = 80)**		**Progression-free survival**				**Overall survival**			
Age (years)		1.00 (0.97–1.03)	0.884			1.01 (0.97–1.05)	0.691		
Female vs. male		1.09 (0.64–1.87)	0.747			1.32 (0.67–2.63)	0.423		
ECOG (≥2 vs. 0.1)		2.27 (1.13–4.56)	0.027	2.27 (1.07–4.80)	0.032	3.48 (1.59–7.63)	0.002	3.76 (1.50–9.42)	0.005
Targetable genetic alteration (yes vs. no)		2.02 (0.85–4.78)	0.110			1.96 (0.75–5.10)	0.171	2.85 (1.03–7.86)	0.043
Lymphocyte count (G/L)		1.08 (0.76–1.53)	0.688			0.84 (0.50–1.41)	0.500		
Squamous-cell vs. adenocarcinoma		1.55 (0.83–2.91)	0.171			1.59 (0.71–3.57)	0.265		
Pack years (≥5 vs. <5)		0.53 (0.27–1.07)	0.075			0.69 (0.26–1.79)	0.685		
CRP (mg/dL)		1.11 (1.06–1.18)	<0.001	1.11 (1.06–1.17)	<0.001	1.16 (1.10–1.22)	<0.001	1.13 (1.07–1.19)	<0.001
PD-L1 status (neg. vs. ≥1%)		1.48 (0.86–2.54)	0.155	1.76 (1.00–3.08)	0.049	1.43 (0.71–2.91)	0.319	1.57 (0.77–3.23)	0.219

HR = hazard ratio, CI= Confidence Interval, ECOG = Eastern Cooperative Oncology Group, CRP = C-reactive protein, PD-L1 = Programmed Death-Ligand 1.

**Table 3 cancers-14-00093-t003:** Uni- and multivariate analyses for progression-free survival in the mono- and chemo-immunotherapy cohort according to sex. Results are presented as hazard ratio (95% confidence interval), with a ratio >1 signifying an increased risk of progression/death.

Progression-Free Survival	Univariate		Multivariate		Univariate		Multivariate	
	HR (95% CI)	*p*	HR (95% CI)	*p*	HR (95% CI)	*p*	HR (95% CI)	*p*
Mono-immunotherapy (*n* = 228)	Male				Female			
Age (years)	0.99 (0.98–1.01)	0.860			0.99 (0.97–1.02)	0.351		
ECOG (≥2 vs. 0.1)	2.19 (1.39–3.45)	<0.001	1.90 (1.10–3.29)	0.021	1.16 (0.71–1.88)	0.553	1.27 (0.74–2.19)	0.384
Therapy line (≥3 vs. 1,2)	1.34 (0.86–2.10)	0.194			1.69 (1.01–2.52)	0.044		
Targetable genetic alteration (yes vs. no)	2.20 (0.88–5.46)	0.091			1.56 (0.83–2.92)	0.165		
Lymphocyte count (G/L)	1.03 (0.90–1.17)	0.660			0.83 (0.62–1.12)	0.228		
Squamous-cell vs. adenocarcinoma	0.99 (0.69–1.43)	0.954			0.92 (0.57–1.51)	0.752		
Pack years (≥5 vs. <5)	0.97 (0.45–2.09)	0.940			0.86 (0.49–1.50)	0.592		
CRP (mg/dL)	1.09 (1.04–1.13)	<0.001	1.06 (1.00–1.11)	0.037	1.01 (0.96–1.07)	0.696		
PD-L1 status (neg. vs. ≥1%)	1.93 (1.26–2.95)	0.002	2.04 (1.32–3.15)	0.001	1.29 (0.79–2.10)	0.316	1.26 (0.78–2.07)	0.344
**Chemo-immunotherapy (*n* = 80)**	**Male**				**Female**			
Age (years)	1.01 (0.97–1.04)	0.717			0.99 (0.96–1.04)	0.936		
ECOG (≥2 vs. 0.1)	2.30 (1.02–5.23)	0.046	1.75 (0.72–4.26)	0.217	6.06 (1.28–28.7)	0.023	5.18 (0.91–29.5)	0.064
Targetable genetic alteration (yes vs. no)	1.33 (0.32–5.61)	0.700			2.24 (0.74–6.84)	0.155		
Lymphocyte count (G/L)	0.97 (0.64–1.48)	0.896			1.60 (0.67–3.78)	0.288		
Squamous-cell vs. adenocarcinoma	2.68 (1.17–6.12)	0.019	4.00 (1.41–11.2)	0.009	1.04 (0.35–3.10)	0.938		
Pack years (≥5 vs. <5)	0.75 (0.29–1.96)	0.556			0.42 (0.15–1.19)	0.101		
CRP (mg/dL)	1.14 (1.04–1.25)	0.004			1.10 (1.04–1.17)	0.001	1.09 (1.02–1.16)	0.007
PD-L1 status (neg. vs. ≥1%)	1.45 (0.72–2.93)	0.297	1.45 (0.70–3.03)	0.321	1.62 (0.68–3.85)	0.274	1.59 (0.65–3.93)	0.311

HR = hazard ratio, CI = Confidence Interval, ECOG = Eastern Cooperative Oncology Group, CRP = C-reactive protein, PD-L1 = Programmed Death-Ligand 1.

**Table 4 cancers-14-00093-t004:** Uni- and multivariate analyses for overall survival in the mono- and chemo-immunotherapy cohort according to sex. Results are presented as hazard ratio (95% confidence interval), with a ratio >1 signifying an increased risk of death.

Overall Survival	Univariate		Multivariate		Univariate		Multivariate	
	HR (95% CI)	*p*	HR (95% CI)	*p*	HR (95% CI)	*p*	HR (95% CI)	*p*
Mono-immunotherapy (*n* = 228)	Male				Female			
Age (years)	1.00 (0.98–1.02)	0.783			0.99 (0.97–1.01)	0.345		
ECOG (≥2 vs. 0.1)	2.44 (1.44–4.12)	<0.001	1.78 (0.97–3.39)	0.063	1.64 (0.96–2.81)	0.072	1.90 (1.04–3.46)	0.037
Therapy line (≥3 vs. 1,2)	1.30 (0.81–2.09)	0.273			1.63 (0.94–2.80)	0.090		
Targetable genetic alteration (yes vs. no)	1.97 (0.79–4.92)	0.145			1.87 (0.97–3.62)	0.060		
Lymphocyte count (G/L)	0.95 (0.78–1.14)	0.567			0.73 (0.51–1.04)	0.726		
Squamous-cell vs. adenocarcinoma	1.11 (0.74–1.66)	0.624			1.25 (0.74–2.10)	0.601		
Pack years (≥5 vs. <5)	0.83 (0.36–1.91)	0.666			1.10 (0.59–2.07)	0.764		
CRP (mg/dL)	1.11 (1.07–1.16)	<0.001	1.09 (1.03–1.14)	0.002	1.05 (0.99–1.10)	0.084		
PD-L1 status (neg. vs. ≥1%)	1.32 (0.83–2.09)	0.245	1.35 (0.84–2.16)	0.216	1.09 (0.64–1.88)	0.747	1.10 (0.64–1.90)	0.738
**Chemo-immunotherapy (*n* = 80)**	**Male**				**Female**			
Age (years)	1.02 (0.96–1.07)	0.535			0.99 (0.95–1.05)	0.956		
ECOG (≥2 vs. 0.1)	4.89 (1.83–13.1)	0.002	5.58 (1.88–16.5)	0.002	3.74 (0.45–31.4)	0.225	1.26 (0.09–16.4)	0.860
Targetable genetic alteration (yes vs. no)	1.09 (0.14–8.30)	0.934			2.16 (0.67–6.68)	0.201		
Lymphocyte count (G/L)	0.62 (0.31–1.26)	0.187			1.63 (0.61–4.35)	0.329		
Squamous-cell vs. adenocarcinoma	1.93 (0.65–5.71)	0.237			1.23 (0.35–4.37)	0.751		
Pack years (≥5 vs. <5)	0.90 (0.21–3.96)	0.886			0.48 (0.13–1.80)	0.277		
CRP (mg/dL)	1.18 (1.08–1.70)	<0.001	1.19 (1.06–1.32)	0.002	1.13 (1.06–1.20)	<0.001	1.11 (1.03–1.19)	0.004
PD-L1 status (neg. vs. ≥1%)	1.27 (0.48–3.34)	0.634	2.78 (0.91–8.46)	0.072	1.78 (0.62–5.09)	0.282	1.39 (0.47–4.13)	0.553

HR = hazard ratio, CI = Confidence Interval, ECOG = Eastern Cooperative Oncology Group, CRP = C-reactive protein, PD-L1 = Programmed Death-Ligand 1.

## Data Availability

According to the terms imposed by the ethics committee, the full datasets analyzed during the current study cannot be made publicly available, as they contain possibly identifiable patient data. Upon reasonable request to the authors and if approved as an amendment by the responsible local ethics committee, selected anonymized data can however be shared.

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
