# Peer review of "Sex-Based Clinical Outcome in Advanced NSCLC Patients Undergoing PD-1/PD-L1 Inhibitor Therapy—A Retrospective Bi-Centric Cohort Study"

_cancers, 2021, doi:10.3390/cancers14010093_

Round 1

Reviewer 1 Report

The authors made a retrospective study to evalute the sex-based and biologic prognostic biomarkers for NSCLC patients undergoing PS-1/PDL1 inhibitor therapy. The study is well conducted, and I hava no comment about the methodology. However, the main outcome is that there are some differences  between male and female for PD-L1 status, ECOG, and histology  prognostic markers value for PFS. There is no noteworthy difference for OS.

These findings are too limited for a full research article, and would be more suitable for a short communication.

Reviewer 2 Report

I would like to congratulate the authors on the completion of this article. It is well presented and the results are well documented.
I suggest these minor revisions:
1. There is an error in table 1 in the age category 70-79 years in the mono-immunotherapy group. 
2. I would like you to specify which variables you have included in the different multivariate analyses.

Reviewer 3 Report

The authors showed a very interesting point in the clinical predictive biomarkers between male and female. This could be relevant for future therapy applicable based on the genre. 

Reviewer 4 Report

Authors reported that men with NSCLC have a more favorable response to immune-checkpoint inhibitor (ICI) monotherapy, while women with NSCLC especially benefit from ICI-chemotherapy (CHT) combinations.

Retrospective analysis to elucidate sex differences in clinical practice of advanced NSCLC using two cohorts treated with either ICI monotherapy (n=228) or ICI-CHT combination treatment (n=80) was shown.

Finally, authors concluded that no significant sex differences for PFS/OS could be detected in either cohort. Although men and women responded similarly to both ICI mono- and ICI-CHT treatment, predictors of response differed by sex.

As authors mentioned in the manuscript, evidence from several metanalyses suggests that men have more favorable responses to ICI monotherapies, while women have a comparably larger benefit on ICI-CHT combinations.

Other previous reports showed that ICIs can improve overall survival for patients with NSCLC, but the magnitude of benefit was sex-dependent.

Authors should consider the previous reports to improve the current manuscript including more female data.

Round 2

Reviewer 1 Report

OK

Reviewer 4 Report

The authors tried to revise the manuscript according to the reviewers' comments. However the author's responce is not sufficient for publication.